# HANSEN: Human and AI Spoken Text Benchmark for Authorship Analysis

**Nafis Irtiza Tripto[1], Adaku Uchendu[1,2], Thai Le[3],**
**Mattia Setzu[4], Fosca Giannotti[4], Dongwon Lee[1]**

[1]The Pennsylvania State University, USA
[1]{nit5154,dongwon}@psu.edu
[2]MIT Lincoln Laboratory, USA
[3]The University of Mississippi, USA ,[4]ISTI-CNR, Pisa, Italy
[2]adaku.uchendu@ll.mit.edu, [3]thaile@olemiss.edu
[4]mattia.setzu@unipi.it,fosca.giannotti@sns.it

## Abstract

*Authorship Analysis*, also known as stylometry, has been an essential aspect of Natural Language Processing (NLP) for a long time. Likewise, the recent advancement of Large Language Models (LLMs) has made authorship analysis increasingly crucial for distinguishing between human-written and AI-generated texts. However, these authorship analysis tasks have primarily been focused on *written texts*, not considering *spoken texts*. Thus, we introduce the largest benchmark for spoken texts - HANSEN (Human ANd ai Spoken tExt beNchmark). HANSEN encompasses meticulous curation of existing speech datasets accompanied by transcripts, alongside the creation of novel AI-generated spoken text datasets. Together, it comprises 17 human datasets, and AI-generated spoken texts created using 3 prominent LLMs: ChatGPT, PaLM2, and Vicuna13B. To evaluate and demonstrate the utility of HANSEN, we perform Authorship Attribution (AA) & Author Verification (AV) on human-spoken datasets and conducted Human vs. AI spoken text detection using state-of-the-art (SOTA) models. While SOTA methods, such as, character n-gram or Transformer-based model, exhibit similar AA & AV performance in human-spoken datasets compared to written ones, there is much room for improvement in AI-generated spoken text detection. The HANSEN benchmark is available at: https://huggingface.co/datasets/HANSEN-REPO/HANSEN.

> **PROMPT**: Create a post match interview with tennis player **Roger Federer** who … <rest of the prompt>. The reporter will ask exactly the following questions: Q:………………………
> *Q: What do you think the Williams sisters' rivalry with one another has meant to the sport?*
>
> **HUMAN**: I don't know. I'd say the players themselves have been great for the sport. I don't know if the matches themselves is what people remember that much.
> **ChatGPT**: The Williams sisters have been amazing for tennis, and their rivalry has been a huge part of that. They've both achieved so much in their careers, and they've pushed each other to be better.
> **PaLM2**: The Williams sisters' rivalry with one another has been one of the most compelling and enduring in tennis history. The rivalry has meant a great deal to the sport of tennis.
> **Vicuna13B**: The Williams sisters' rivalry has been incredible for the sport. It's been amazing to see the two of them compete against each other and push each other to be their best.

Figure 1: Snippet of the HANSEN AI-text samples. Given the actual question (part of the prompt), the HUMAN (ROGER FEDERER) answer vs three LLMs: ChatGPT, PaLM2, and Vicuna13B.

## 1 Introduction

Authorship analysis is a longstanding research area in NLP that has garnered significant attention over the years. Two tasks are at the core of authorship analysis - Authorship Attribution (AA) and Authorship Verification (AV). AA is the process of identifying the author of a text document (Kjell et al., 1994). Similarly, AV determines whether two documents are written by the same author (Halteren, 2007). Both have been extensively researched as text classification problems due to their substantial impacts on various applications, such as author profiling, author forensic analysis, resolving copyright disputes, and similar issues (Neal et al., 2017). Currently, most of the research in NLP text classification focuses on written text due to vast amounts of text data available for training and evaluation (Kowsari et al., 2019). However, "text" is also inherent in spoken language as a textual representation of what individuals say, commonly known as *spoken text* (Biber, 1991). Although spoken language has always existed before written language (i.e., considering human history of language as a twelve-inch ruler, written language has only existed for the *"last quarter of an inch"* (Wrench et al., 2008)), it has not received much attention from the NLP community. Simultaneously, recent advancements in speech-to-text technology have

| Written example | Spoken example | Remarks |
|---|---|---|
| Today, the Commission considers adopting a final rule to enhance the disclosures related to share buybacks. I support this final rule because it will increase[...] First, the rule will require issuers to disclose periodically the prior period's daily buyback activity. This will include such information as the date of the purchase, the amount of shares repurchased, and the average purchase price for the date. [...] Second, issuers are required by the rule to provide disclosure about their buyback programs. Such disclosure will include details about the objectives or rationales for the buyback as well as the process[...] | Good morning. I am pleased to join the Investor Advisory Committee. [...] I look forward to hearing about your potential recommendation today regarding customer account statements. We look forward to your input about private markets,[...] We also look forward to your discussion on open-end funds. Now, let me turn to your panel on the oversight of investment advisers. I am glad you are discussing advisers, because [...] | Written statement and spoken speech from US Security Exchange (SEC) Commission Chairperson Gary Gensler about corresponding issues. |
| *Different people naturally have differing life experiences and differing viewpoints – which inevitably results in varying agendas when implementing the law according to their own personal discretion. In this respect – Birks is correct that an over-reliance on conscience alone to provide equitable solutions inevitably leads to a messy framework of case law that would result in judges navigating uncertainly by feel than by the solid path of precedent. ..* | So I do want to try steak, yeah. I think it's[...] I-I don't think it's bad, I think it's a good thing. Erm, because I-I did try be veggie for, I think it was six months I did it, but I gained a lot of weight. I think probably because I didn't really know the other options I could have ate instead. I think I was eating more carbs and stuff like that. So I was gaining weight so I stopped[...] | Written essay and informal interview transcript from a sample participant in Aston Idiolects Corpus. |

Figure 2: Written and spoken text samples from same individual. Written texts contain linking words, passive voice (Akinnaso, 1982), complicated words, and *complex sentence structures*. However, spoken texts contain more first & second person usages, informal words (Brown et al., 1983), grammatically incorrect phrases (Biber et al., 2000), repetitions, and other differences. Sentences in spoken texts are also shorter (Farahani et al., 2020).

expanded the availability of spoken text corpora, enabling researchers to explore new avenues for NLP research focused on spoken language.

Identification of speakers from speech has been successfully addressed through various audio features, with or without the availability of associated text information (Kabir et al., 2021). Whether speaker identification can be achieved solely based on how individuals speak, as presented in spoken text, remains mostly unexplored in the existing literature. However, the rise of audio podcasts and short videos in the digital era, driven by widespread social media usage, has increased the risk of plagiarism as individuals seek to imitate the style and content of popular streamers. Therefore, spoken text authorship analysis can serve as a valuable tool in addressing this situation. It can be challenging for several reasons. First, text is not the only message we portray in speaking since body language, tone, and delivery also determine what we want to share (Berkun, 2009). Second, spoken text is more casual, informal, and repetitive than written text (Farahani et al., 2020) Third. the word choice and sentence structure also differ for spoken text (Biber, 1991). For example, Figure 2 portrays the difference between written and spoken text from the same individuals. Therefore, discovering the author's style from the spoken text can be an exciting study leveraging the current advanced AA and AV techniques.

On top of classical plagiarism detection problems, there is a looming threat of synthetically generated content from LLMs. Therefore, using LLMs, such as ChatGPT, for generating scripts for speech,

podcasts, and YouTube videos is expected to become increasingly prevalent. Figure 1 exemplifies how various LLMs can adeptly generate replies to an interview question. While substantial research efforts are dedicated to discerning between human and AI-generated text, also known as Turing Test (TT) (Uchendu et al., 2023), evaluating these methods predominantly occurs in the context of written texts, such as the Xsum (Narayan et al., 2018) or SQuAD (Rajpurkar et al., 2016) datasets. Thus, spoken text authorship analysis and detecting AI-generated scripts will be crucial for identifying plagiarism & disputing copyright issues in the future.

To tackle these problems, we present HANSEN (Human ANd ai Spoken tExt beNchmark), a benchmark of human-spoken & AI-generated spoken text datasets, and perform three authorship analysis tasks to better understand the limitations of existing solutions on "spoken" texts. In summary, our contributions are as follows.

- We compile & curate existing speech corpora and create new datasets, combining 17 human-spoken datasets for the HANSEN benchmark.
- We generate spoken texts from three popular, recently developed LLMs: ChatGPT, PaLM2 (Anil et al., 2023), and Vicuna-13B (Chiang et al., 2023) (a fine-tuned version of LLaMA (Touvron et al., 2023) on user-shared conversations, combining ~21K samples in total, and evaluate their generated spoken text quality.
- We assess the efficacy of traditional authorship analysis methods (AA, AV, & TT) on these HANSEN benchmark datasets.

## 2 Related Work

**Authorship analysis:** Over the last few decades, both AA and AV have been intensively studied with a wide range of features and classifiers. N-gram representations have been the most common feature vector for text documents for a long time (Kjell et al., 1994). Various stylometry features, such as lexical, syntactic, semantic, and structural are also popular in authorship analysis (Stamatatos, 2009; Neal et al., 2017). While machine learning (ML) & deep learning (DL) algorithms with n-gram or different stylometry feature sets have been popular in stylometry for a long time (Neal et al., 2017), Transformers have become the state-of-the-art (SOTA) model for AA and AV, as well as many other text classification tasks (Tyo et al., 2022). Transformers, such as BERT (Bidirectional Encoder Representations from Transformers) (Devlin et al., 2018) and GPT (Generative Pre-trained Transformer) (Radford et al., 2019), are generally pre-trained on large amounts of text data before fine-tuning for specific tasks (Wang et al., 2020), such as AA & AV. Also, different word embeddings, such as GLOVE (Pennington et al., 2014) and FastText (Bojanowski et al., 2017), can capture the semantic meaning between words and are employed with DL algorithms for AA as well.

**LLM text detection:** LLMs' widespread adoption and mainstream popularity have propelled research efforts toward detecting AI-generated text. Supervised detectors work by fine-tuning language models on specific datasets of both human & AI-generated texts, such as the GPT2 detector (Solaiman et al., 2019), Grover detector (Zellers et al., 2019), ChatGPT detector (Guo et al., 2023), and others. However, they only perform well within the specific domain & LLMs for which they are fine-tuned (Uchendu et al., 2023; Pu et al., 2023). Statistical or Zero-shot detectors can detect AI text without seeing previous examples, making them more flexible. Therefore, it has led to the development of several detectors, including GLTR (Gehrmann et al., 2019), DetectGPT (Mitchell et al., 2023), GPT-zero (Tian, 2023), and watermarking-based techniques (Kirchenbauer et al., 2023).

**Style from speech:** While some studies have attempted to infer the author's style from speech, they primarily focus on the speech emotion recognition task and rely on audio data. These works (Shinozaki et al., 2009; Wongpatikaseree et al., 2022)

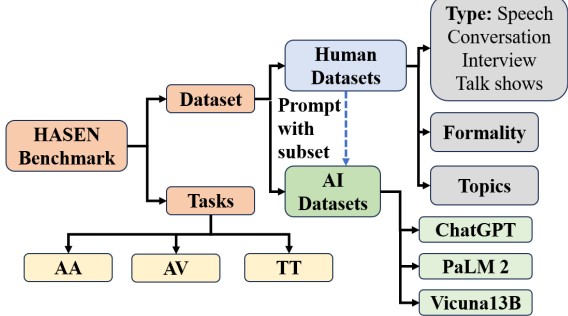

Figure 3: HANSEN benchmark framework

utilize various speech features, and DL approaches to recognize the speaker's emotions, such as anger or sadness, and subsequently define the speaking style based on these emotional cues.

## 3 Building the Benchmark: HANSEN

Existing conversational toolkits and datasets, such as ConvoKit (Chang et al., 2020), prioritize analyzing social phenomena in conversations, thus limiting their applicability in authorship analysis. Similarly, current LLM conversational benchmarks, including HC3 (Guo et al., 2023), ChatAlpaca (Bian et al., 2023), and XP3 (Muennighoff et al., 2023), primarily focus on evaluating LLMs' question-answering and instruction-following abilities rather than their capability to generate authentic spoken language in conversations or speeches. Therefore, we introduce the HANSEN datasets to address this gap, incorporating human and AI-generated spoken text from different scenarios as portrayed in Figure 3, thereby providing a valuable resource for authorship analysis tasks. The datasets of HANSEN are available through Python Hugging Face library.

### 3.1 Human *spoken text* datasets

The HANSEN benchmark comprises 17 human datasets, where we utilize several existing speech corpora and create new datasets. Our contribution involves curating and preparing these datasets for authorship analysis by leveraging metadata information, aligning spoken text samples with their respective speakers, and performing necessary post-processing. Table 1 shows the summary of the human datasets.

**Existing datasets selection:** When constructing the HANSEN benchmark datasets, several criteria guided our selection of speech corpora. Firstly, we consider datasets with readily available transcripts (e.g., TED, BNC, BASE) or the ability to

| Dataset | Type Topic | Description | Characteristics | Sample Definition | #Speakers | #Samples | #Tokens |
|---|---|---|---|---|---|---|---|
| **TED** -talk | Speech Mixed | Transcripts of TED talks collected from the official website. | Mostly unscripted & informal | Entire speech | 3349 | 4122 | 7.19M |
| **Spotify** | Speech Mixed | It is a collection of English-language Spotify podcast episodes transcripts compiled by (Clifton et al., 2020). | Mostly unscripted & informal | Entire speech | 17490 | 105357 | 684.99M |
| **BASE** | Speech/ Conversation Academic | The British Academic Spoken English (BASE) corpus (Thompson and Nesi, 2001) comprises lectures and seminars recorded in a university. | Mostly unscripted & informal discussion | All utterance in a session lecture/ seminar) | 523 | 3077 | 1.66M |
| **BNC** | Conversation Daily life topics | We utilize the spoken portion of the British National Corpus (BNC) (Consortium et al., 2007). It is a collection of late-twentieth century British English language. | Unscripted informal conversations | All utterance in a conversation | 2461 | 90153 | 53.6M |
| **BNC14** | Conversation Daily life topics | A contemporary version of the previous corpus (Love et al., 2017). Both datasets are gathered from various real-life contexts. | Unscripted & informal | All utterance in a conversation | 459 | 1868 | 4.65M |
| **MSU** Switchboard | Conversation pre-defined topics | The MSU Switchboard Dialog Act Corpus (Stolcke et al., 2000) includes phone calls between two participants. The callers ask receivers questions about child care, recycling, the news media, and other provided topics. | Unscripted & informal | All utterance in a phone call | 440 | 2310 | 1.91M |
| **PAN23** (Aston Idiolect corpus) | Speech/ conversation Daily life topics | It is the training set of PAN-23 AV task and originally introduced by (Petyko et al., 2022). The spoken part contains interviews and speech transcriptions from students. | Unscripted & informal | All utterance in a session | 56 | 17672 | 14.44M |
| **Tennis** | Interview Sports | Originally introduced by (Liye et al., 2016), it contains Tennis single post-match interviews in major tournaments 2007-2015. | Unscripted & informal | All answers in an interview | 358 | 6467 | 6.58M |
| **CEO** interview | Interview Financial | It contains interviews with the CEO of companies and other financial persons associated with stock markets. We created it from the available public transcripts of these interviews from three different sources: CEO-today magazine, Wall Street Journal, and Seeking Alpha website. | Unscripted & formal | Each individual answer since very few interview available for each CEO | 6298 | 15072 | 7.01M |
| **Voxceleb** | Interview Celebrity | It comprises interview transcripts featuring various celebrities sourced from YouTube. Leveraging the original corpus established by (Nagrani et al.)., we extract the YouTube transcripts to form this version. | Unscripted & informal | All answers in an interview | 3753 | 57702 | 16.05M |
| British Parliament **(BP)** | Question/ Answer Politics | Initially introduced by (Zhang et al., 2017), this dataset contains questions & answers from the British House of Commons. | Scripted/ Unscripted & formal | All utterance in a session | 1065 | 23849 | 18.66M |
| **Voxpopuli** | Speech Politics | We utilize the English portion of the original Voxpopuli (Wang et al., 2021) dataset, which contains the raw data from European Parliament recordings. | Scripted/ Unscripted & formal | All utterance in a session | 924 | 24549 | 73.1M |
| Face the Nation **(FTN)** | Talk-show Mixed | We created this dataset by compiling the transcripts from the popular talk show, Face the Nation by CBS News. Each episode is led by a host who discusses contemporary topics with multiple guests. | Mostly unscripted & formal | Each utterance | 1613 | 84942 | 5.24M |
| US Life Podcast **(USP)** | Talk-show Mixed | Initially introduced by (Mao et al., 2020) it contains podcasts from the US Life radio program from 1995 to 2020. | Unscripted & informal | All utterance from an Act in episode | 165 | 3881 | 1.51M |
| **SEC** | Speech Financial | We compile this dataset from the Security Exchange Commission (SEC) press release. It contains the transcripts of speech (spoken text) and statements (written text) from various personnel. | Scripted & formal | Entire speech | 59 | 1101 | 1.46M |
| **Debate** | Argumentative conversation Mixed | This dataset, introduced by (Zhang et al., 2016) contains transcripts of debates held as part of Intelligence Squared Debates. | Can contain both characteristics | Each utterance | 470 | 5766 | 1.82M |
| Supreme **Court** | Argumentative conversation Legal | This massive corpus consists of a collection of cases from the U.S. Supreme Court, along with transcripts of oral arguments. The transcripts were collected from the Oyez website (Urofsky, 2001) | Mostly unscripted & formal | All utterance in a session | 8978 | 66758 | 79.66M |

Table 1: Summary of the HANSEN Human datasets.

extract transcripts (e.g., Voxceleb (Nagrani et al.) via YouTube automated transcriptions). Secondly, speaker information for each sample was required, leading to the exclusion of corpora such as Coca (Davies, 2015). Finally, we also emphasize datasets where speakers did not recite identical content or scripts, excluding datasets like Ljspeech (Ito and Johnson, 2017) and Movie-Dialogs Corpus (Danescu-Niculescu-Mizil and Lee, 2011) for their lack of spontaneity & originality in representing natural spoken language.

**New dataset creation:** We have introduced three new datasets within our benchmark[1]. The **SEC** dataset comprises transcripts from the Security Exchange Commission (SEC) website, encompassing both spoken (speech) and written (statements) content from various personnel in the commission. The Face the Nation (**FTN**) dataset compiles transcripts from the Face the Nation talk show by CBS News, where episodes feature discussions on contemporary topics with multiple guests. The **CEO** interview dataset contains interviews with financial personnel, CEOs of companies, and stock market associates assembled from public transcripts provided by CEO-today magazine, Wall Street Journal, and Seeking Alpha. For authorship analysis, we retained only the answers from the interviewees, excluding the host's questions.

**Dataset curating:** We have followed a systematic process for all datasets in our benchmark preparation. It involves eliminating tags, URLs, & metadata from the text, and aligning each utterance with its respective speaker for generating labeled data for authorship analysis. To maintain consistent text lengths and contextual coherence within utterances, we adjust the span of each sample, as detailed in Table 1. In cases where a sample was excessively long, such as an lengthy TED talk, we split it into smaller segments to ensure uniform lengths. Furthermore, we partition each dataset into training, validation, and testing subsets while preserving a consistent author ratio across these partitions.

| LLM | TED | Spotify | SEC | CEO | Tennis | Total |
|---|---|---|---|---|---|---|
| **ChatGPT** | 3491 | 2778 | 301 | 61 | 2021 | 8652 |
| **PaLM2** | 2461 | 2040 | 284 | 59 | 1991 | 6835 |
| **Vicuna13B** | 2868 | 2524 | 290 | 60 | 2008 | 7750 |

Table 2: LLM-generated samples in each category

---

[1] link of the websites & other details in Appendix

**Dataset characteristics:** The benchmark is representative of a diverse range of naturally occurring dialogue forms, encompassing monologue (speech/podcast/lecture), dialogue (conversation/interview/argument), and multi-party dialogue (talk show). It also includes scripted and unscripted speech across a spectrum of formality levels. The transcripts of these datasets are often manually generated and annotated as part of the official corpora (e.g., BASE, BNC, BNC14, MSU), obtained directly from official sources (e.g., TED, USP, Tennis) or automatically generated through speech to text techniques (e.g., Spotify & Voxpopuli). Given the topic-dependent nature of authorship analysis (Sari et al., 2018), the benchmark comprises datasets covering specific or diverse topics.

## 3.2 AI (LLM)-generated *spoken text* dataset

With the growing societal impact of LLMs, we will soon observe their extensive use in generating scripts for speech and guidelines for interviews or conversations. Thus, it motivates us to include AI-generated spoken text in HANSEN. Since chat-based LLMs can follow instructions and generate more conversation-like text than traditional LLMs (Ouyang et al., 2022), we have utilized three recent prominent chat-based LLMs: **ChatGPT** (gpt3.5-turbo), **PaLM2** (chat-bison@001), and **Vicuna-13B** in our study. We also ensure the spoken nature of the generated texts and evaluate their quality.

***Spoken* text generation:** LLMs are predominantly trained on written text from diverse sources, including BookCorpus (Zhu et al., 2015), Open WebText (Aaron Gokaslan*, 2019), and Wikipedia (Merity et al., 2017), containing a small portion of spoken texts, while the exact proportion is unknown. Therefore, effective prompt engineering is crucial for generating *spoken text*.

Different metadata associated with each speaker and text samples can be utilized to construct efficient prompts. For instance, by leveraging elements like talk descriptions, speaker details, and talk summaries, LLMs can produce more coherent TED talk samples rather than providing the topic only. To create spoken texts from LLMs, we utilize subsets from five human datasets (**TED**, **SEC**, **Spotify**, **CEO**, & **Tennis**), chosen for their rich metadata and the involvement of notable public figures as speakers. Additional insights regarding prompt construction, dataset selection, and related attributes can be found in the Appendix.

| Datasets | SEC | | PAN | | ChatGPT | | PaLM2 | | Vicuna13B | |
|---|---|---|---|---|---|---|---|---|---|---|
| Features | written | spoken | written | spoken | written | spoken | written | spoken | written | spoken |
| Avg word length | 5.50±0.31 | 5.30±0.29 | 5.56±0.46 | 4.31±0.17 | 5.19±0.44 | 4.93±0.41 | 4.90±0.53 | 4.78±0.47 | 5.11±0.43 | 4.84±0.42 |
| Avg word in sen | 25.45±4.44 | 22.16±3.82 | 22.11±5.77 | 17.87±2.75 | 20.17±2.73 | 16.85±3.60 | 17.14±5.33 | 14.74±2.95 | 21.80±3.81 | 18.02±3.55 |
| Voc. richness | 0.49±0.06 | 0.51±0.05 | 0.40±0.09 | 0.32±0.04 | 0.45±0.05 | 0.45±0.08 | 0.45±0.14 | 0.38±0.07 | 0.51±0.10 | 0.43±0.07 |
| Long word fraction | 0.29±0.04 | 0.26±0.03 | 0.22±0.07 | 0.10±0.02 | 0.26±0.07 | 0.22±0.07 | 0.22±0.08 | 0.20±0.08 | 0.26±0.07 | 0.21±0.07 |
| 1st person count | 6.98±5.90 | 8.06±5.54 | 9.82±11.84 | 28.70±14.66 | 3.39±5.77 | 14.75±11.09 | 5.39±7.47 | 19.04±12.24 | 2.77±5.74 | 15.34±10.95 |
| 2nd person count | 0.76±1.80 | 1.83±2.83 | 4.92±6.16 | 5.37±6.43 | 1.75±4.19 | 6.85±5.87 | 4.07±8.79 | 9.60±8.20 | 1.00±2.67 | 6.34±5.73 |
| Readability score | 36.89±11.37 | 45.73±9.74 | 50.11±15.54 | 80.39±6.27 | 51.94±14.48 | 62.26±15.39 | 63.99±19.22 | 68.53±16.61 | 51.13±15.82 | 62.67±15.47 |
| Text DIV score | 1.19±0.17 | 1.13±0.19 | 1.12±0.11 | 1.10±0.11 | 1.17±0.26 | 1.12±0.15 | 1.25±0.51 | 1.00±0.20 | 1.34±0.34 | 1.12±0.21 |
| Misspelling % | 12.50±2.89 | 13.37±2.44 | 13.96±3.61 | 16.64±2.89 | 10.04±2.03 | 11.36±2.37 | 10.49±1.93 | 11.75±6.29 | 10.67±3.05 | 10.86±2.77 |

Table 3: Feature difference between written vs spoken text. The value in corresponding cell indicate the *mean±variance* of that feature. Red colored cell indicate statistical significance was **not found** for that feature.

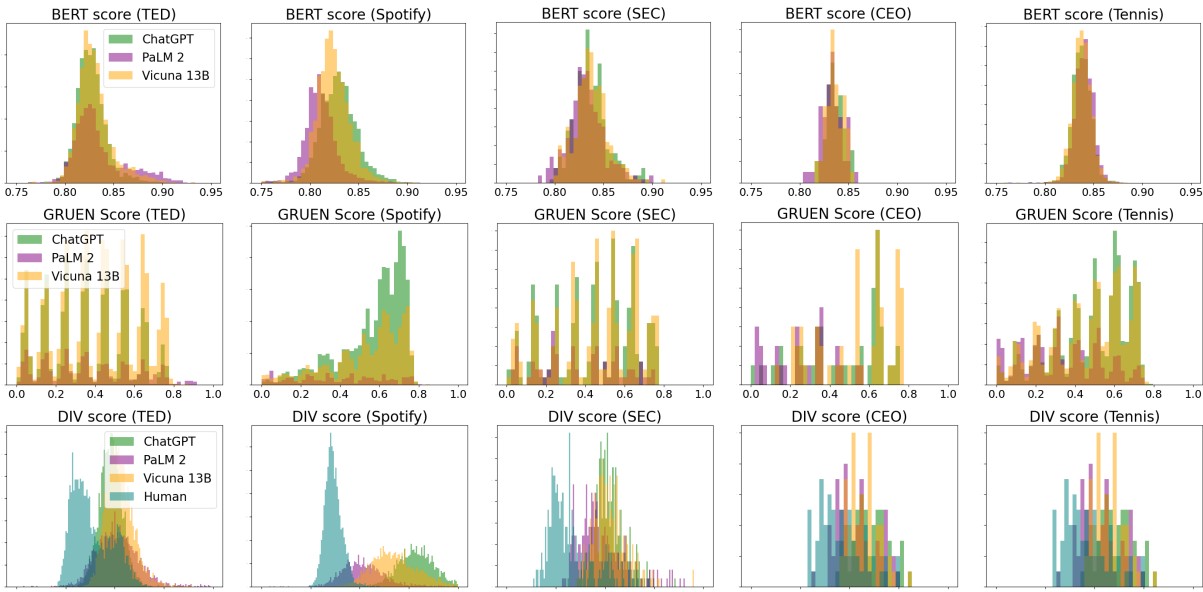

Figure 4: Distribution of BERT, GRUEN, Text diversity (DIV) score for three LLMs (the higher, the better).

| Dataset | TED (Speech) | | | Spotify (Speech) | | | SEC (Speech) | | | CEO (Interview) | | | Tennis (Interview) | | |
|---|---|---|---|---|---|---|---|---|---|---|---|---|---|---|---|
| # of speakers | 10 | 10(L) | 50 | 10 | 10(L) | 100 | 10 | 10(L) | 30 | 10 | 10(L) | 100 | 10 | 10(L) | 100 |
| **Char n-gram** | **0.95** | **0.89** | **0.93** | **0.98** | 0.91 | **0.82** | 0.9 | 0.9 | 0.54 | 0.8 | 0.86 | 0.58 | 0.99 | 0.83 | **0.84** |
| **Stylometry** | 0.71 | 0.67 | 0.64 | 0.87 | 0.83 | 0.61 | 0.68 | 0.69 | 0.46 | 0.6 | 0.68 | 0.35 | 0.91 | 0.89 | 0.66 |
| **FastText** | 0.73 | 0.59 | 0.6 | 0.93 | 0.84 | 0.47 | 0.68 | 0.67 | 0.42 | 0.59 | 0.61 | 0.28 | 0.87 | 0.87 | 0.46 |
| **Bert-AA** | 0.69 | 0.53 | 0.48 | 0.9 | 0.87 | 0.54 | 0.55 | 0.64 | 0.22 | 0.72 | 0.74 | 0.55 | 0.76 | 0.84 | 0.22 |
| **BERT-ft** | 0.89 | 0.72 | 0.75 | **0.98** | **0.94** | 0.78 | 0.83 | 0.81 | **0.63** | 0.76 | 0.83 | **0.62** | 0.91 | **0.94** | 0.59 |

Table 4: Macro-F1 score for different AA-methods for variable number of speakers. 10(L) indicates the setup where each LLM has been considered as a speaker (7 human speakers with 3 LLMs).

**Evaluating *spoken* nature of AI-generated texts:** To ensure the *spoken* nature of AI-generated texts, corresponding *written* texts in the same context are essential. We select subsets for each category and instruct LLMs to generate written articles based on their respective spoken texts. In HANSEN, only two human datasets (SEC and PAN) provide written and spoken samples from the same individuals. Stylometry properties of these samples from human and LLM-generated datasets are compared, and a student t-test (Livingston, 2004) was used to determine if there were significant differences in these stylometric features between written and

spoken texts. The results (Table 3) indicate that LLM-generated spoken and written text exhibits similar trends to human datasets, including variations in word lengths, first/second person counts, vocabulary richness, and other linguistic features, thus validating that LLMs have successfully generated spoken text with a spoken-like quality in most scenarios.

ChatGPT and Vicuna13B demonstrate similar differences compared to humans, whereas PaLM 2 shows minor differences in some cases, indicating the need for more training with human-spoken texts. It is essential to consider the distinct nature

of the SEC and PAN datasets; SEC speeches and statements are rigorously scripted and reviewed due to their potential impact on financial markets and the economy (Burgoon et al., 2016), while the PAN dataset comprises mainly informal spoken and written texts. Additional detailed findings are available in the Appendix.

**Quality of AI-generated spoken text:** Additionally, we use different automatic evaluation metrics, such as BERT score (Zhang* et al., 2020), GRUEN (Zhu and Bhat, 2020), MAUVE (Pillutla et al., 2021), text diversity score, and readability score (Kincaid et al., 1975) to evaluate quality of the generated text (Figure 4). In interview datasets, the BERT scores for all LLMs are consistently higher than 0.8 (out of 1.0) and mostly similar, as the original questions guide the generated texts. However, for open-ended speech datasets like TED and Spotify, PaLM2 shows lower BERT scores compared to other models. The GRUEN scores across all datasets are uniformly distributed, indicating variation in linguistic quality among individual-generated samples. PaLM2 exhibits lower GRUEN scores than the other models. Surprisingly, we observe lower text diversity for human texts, contradicting existing AI vs. human analyses (Guo et al., 2023) but supporting the repetitive nature of human-spoken texts (Farahani et al., 2020).

## 4   Authorship Analysis on HANSEN

We conduct three authorship analysis tasks using HANSEN and evaluate how existing SOTA methods perform in the context of "spoken" texts. Details on AA, AV, & TT methods are provided in Tables 13, 14, 15 in the Appendix section. We run the experiments five times and report the average.

### 4.1   Author Attribution (AA)

Author Attribution (AA) is a closed-set multi class classification problem that, given a spoken text $T$, identifies the speaker from a list of candidate speakers. We have primarily performed AA on human datasets as well as considering each LLM as an individual speaker. We have utilized N-grams (character & word), Stylometry features (WritePrint (Abbasi and Chen, 2008) + LIWC (Pennebaker et al., 2001)), FastText word embeddings with LSTM, BERT-AA (Fabien et al., 2020), Finetuned BERT (BERT-ft) as AA methods.

Additionally, we have evaluated the realistic scenario where each LLM performs as an individual

| Dataset | TED | | SEC | | Tennis | |
|---|---|---|---|---|---|---|
| # of speakers | 20 | 20(L) | 20 | 20(L) | 20 | 20(L) |
| Char n-gram | 0.92 | 0.37 | 0.7 | 0.33 | 0.95 | 0.42 |
| Stylometry | 0.68 | 0.39 | 0.55 | 0.29 | 0.65 | 0.43 |
| BERT-ft | 0.84 | 0.51 | 0.73 | 0.45 | 0.84 | 0.67 |

Table 5: Ablation study comparing AA-20(L) as top 5 human speakers and corresponding samples from 3 LLMs (considering as 20 distinct classes) against a scenario with only the top 20 human speakers. Results show a substantial performance drop, indicating LLMs' ability to impersonate human speakers.

speaker and performs AA on ten speakers (7 humans + 3 LLMs). Also, texts from the same LLM are generated using a specific prompt (prompting them to symbolize the actual human persona). Thus, LLM is supposed to exhibit a different persona in all its instances. Therefore, we have performed an ablation study using samples from the top 5 human speakers (with most samples) and their corresponding samples generated by ChatGPT, Vicuna13B, and PaLM2, resulting in a total of 20 classes for comparison. We employed conventional text classification evaluation metrics, including Accuracy, macro F1 score, Precision, Recall, and Area under the curve (AUC) score. Table 4, 5 and 6 present the macro F1 score and demonstrate that character (char) n-gram performs best in most scenarios, with BERT-ft being a close contender. Our observations align with the AA findings of Tyo et al. (2022) on written text datasets. However, the performance of BERT-AA is subpar when directly applied to spoken text datasets, but finetuning them improves performance substantially.

### 4.2   Author Verification (AV)

Author verification (AV) is a binary classification problem that, given a pair of spoken texts $(T_1, T_2)$, detects whether they were generated by the same speakers or different speakers. We have used PAN AV baselines: N-gram similarity, Predictability via Partial Matching (PPM), and current PAN SOTA methods: Adhominem (Boenninghoff et al., 2019), Stylometry feature differences (Weerasinghe et al., 2021), and finetuned BERT. We have used tradiional PAN evaluation metrics for AV (Bevendorff et al., 2022), including F1, AUC, c@1, F_0.5u, and Brier scores. Table 7 reports the F1 score for several datasets. Unlike AA, Adhominem showed the best performance in AV, with finetuned BERT being particularly notable.

| Type/Topic | Conversation (daily-life topic/other) | | | | | Interview | Speech (Political) | | Arguments | | Talk shows | | |
|---|---|---|---|---|---|---|---|---|---|---|---|---|---|
| Dataset | BASE | BNC | BNC14 | MSU | PAN | Voxceleb | BP | Voxpopuli | Court | Debate | USP | FTN | AVG |
| Char n-gram | 0.98 | **0.89** | **0.92** | 0.33 | 1 | 0.86 | **0.98** | **0.94** | **0.91** | 0.83 | **0.52** | 0.58 | **0.84** |
| Stylometry | 0.84 | 0.71 | 0.79 | 0.24 | 1 | 0.68 | 0.84 | 0.76 | 0.79 | 0.74 | 0.42 | 0.42 | 0.70 |
| FastText | 0.8 | 0.64 | 0.76 | **0.33** | 1 | 0.77 | 0.84 | 0.79 | 0.73 | 0.58 | 0.28 | 0.51 | 0.69 |
| Bert-AA | 0.94 | 0.7 | 0.66 | 0.26 | 1 | **0.87** | 0.69 | 0.87 | 0.82 | 0.72 | 0.3 | 0.64 | 0.71 |
| BERT-ft | **0.99** | 0.88 | 0.9 | 0.28 | 1 | **0.87** | 0.92 | 0.93 | 0.89 | **0.85** | 0.4 | **0.65** | 0.82 |

**Bold** and underlined values represent each dataset's highest and second-highest-performing method (based on macro F1).

Table 6: AA result for N=10 speakers in different datasets. Results with higher value of N are present in Appendix.

| Type/Topic | Speech | | | Conversation | | | Interview | | Political | Legal | Talk shows | | |
|---|---|---|---|---|---|---|---|---|---|---|---|---|---|
| Dataset | TED | Spotify | SEC | BASE | MSU | PAN | CEO | Tennis | Voxpopuli | Court | USP | FTN | AVG |
| Char n-gram | 0.86 | 0.67 | 0.67 | 0.78 | **0.66** | 0.67 | 0.68 | 0.68 | 0.64 | 0.61 | 0.65 | 0.66 | 0.68 |
| PPM | 0.85 | 0.65 | 0.63 | 0.76 | 0.57 | 0.55 | 0.67 | 0.72 | 0.61 | 0.64 | 0.62 | 0.55 | 0.66 |
| Feature Diff. | 0.87 | 0.69 | 0.6 | 0.72 | 0.47 | 0.99 | 0.72 | 0.76 | 0.55 | 0.73 | 0.67 | 0.51 | 0.7 |
| Adhominem | 0.84 | **0.88** | **0.81** | **0.96** | 0.62 | **1.0** | **0.93** | **0.91** | **0.8** | **0.91** | **0.92** | **0.72** | **0.87** |
| BERT-ft | **0.91** | 0.83 | 0.64 | 0.93 | 0.46 | 0.67 | 0.88 | 0.81 | 0.73 | 0.79 | 0.81 | 0.67 | 0.77 |

Table 7: AV results (F1 score) for several datasets. Results for other metrics and datasets are presented in Appendix.

| LLM | TED | Spotify | SEC | CEO | Tennis |
|---|---|---|---|---|---|
| **ChatGPT** | M: 0.59 | M: 0.53 | M: 0.51 | M: 0.69 | M: 0.51 |
| | $\overline{R}$: 0.82 | $\overline{R}$: 0.7 | $\overline{R}$: 0.74 | $\overline{R}$: 0.76 | $\overline{R}$: 0.82 |
| **PaLM 2** | M: 0.82 | M: 0.65 | M: 0.71 | M: 0.82 | M: 0.59 |
| | $\overline{R}$: 0.88 | $\overline{R}$: 0.87 | $\overline{R}$: 0.81 | $\overline{R}$: 0.96 | $\overline{R}$: 0.97 |
| **Vicuna 13B** | M: 0.74 | M: 0.58 | M: 0.68 | M: 0.75 | M: 0.58 |
| | $\overline{R}$: 0.83 | $\overline{R}$: 0.8 | $\overline{R}$: 0.8 | $\overline{R}$: 0.85 | $\overline{R}$: 0.93 |

Table 8: The Mauve score (M) and avg recall value ($\overline{R}$) (considering all detectors) for each LLM. A higher **M** means the distribution is more similar to humans. A higher **$\overline{R}$** means that the LLM is easily detectable. The **violet color** indicates better performance for an LLM (higher Mauve score and lower recall value), while the red indicates worse performance (and vice versa).

| Methods | SEC(w) | SEC(s) | PAN(w) | PAN(s) |
|---|---|---|---|---|
| Char N-gram | F1: 0.58 | F1: 0.57 | F1: 0.45 | F1: 0.41 |
| | $\Delta$:(-0.31) | $\Delta$: (-0.32) | $\Delta$: (-0.55) | $\Delta$: (-0.59) |
| Stylometry | F1: 0.38 | F1: 0.41 | F1: 0.18 | F1: 0.21 |
| | $\Delta$: (-0.30) | $\Delta$: (-0.27) | $\Delta$: (-0.82) | $\Delta$: (-0.79) |
| Finetuned BERT | F1: 0.5 | F1: 0.53 | F1: 0.32 | F1: 0.35 |
| | $\Delta$: (-0.33) | $\Delta$: (-0.30) | $\Delta$: (-0.68) | $\Delta$: (-0.65) |

Table 9: Results on combining both spoken & written samples from same individuals. SEC/PAN (w) specifies that the training set was written only and test set was spoken texts by same speakers. SEC/PAN (s) specifies the vice-versa. Each cell value is the macro F1 score and the difference in F1 score ($\Delta$) with the original PAN/SEC dataset in the AA-10 class problem.

## 4.3 Turing Test (TT) for Spoken Text

We frame the human vs AI spoken text detection problem as *Turing Test (TT)*, a binary classification problem that, given a spoken text $T$, identifies whether it is from a human or AI (LLM). We have utilized several supervised and zero-shot detectors: OpenAI detector, Roberta-Large, DetectGPT, and GPT Zero in our study. Table 10 highlights the results of different TT methods in various datasets/LLMs. Notably, no single method emerges as the apparent "best" option, as performance varies substantially across different datasets. Overall, the OpenAI text detector excels in speech datasets (TED, Spotify, SEC), while DetecGPT performs better in interview datasets (CEO, Tennis). However, all methods exhibit limitations across various settings, with either low precision or low recall. For instance, GPT Zero demonstrates low precision scores in interview datasets and TED talks, suggesting perplexity and burstiness measures may

vary in spoken text. Furthermore, the low recall of Roberta-Large, specifically for ChatGPT, may be attributed to the distinct nature of spoken texts generated by ChatGPT compared to the written texts from previous GPT models.

## 5 Discussion

**Character n-gram dominates in AA but struggles in AV:** While character n-gram performs best in AA for most datasets, it underperforms in the AV task. More intricate DL models, such as Adhominem, excel in AV, consistent with findings in written text datasets (Tyo et al., 2022). Larger DL models tend to outperform smaller models when the datasets have more words per class, leading to better AV performance since it only has two classes (Tyo et al., 2022). Character n-grams outperform word n-grams by a notable margin (5%-10% in general), emphasizing the potential enhancement of AA performance through the inclusion of informal words (e.g., "eh," "err," "uhh").

| Dataset | TED (S, mixed) | | | Spotify (S, mixed) | | | SEC (S, financial) | | | CEO (I, financial) | | | Tennis (I, sports) | | |
|---|---|---|---|---|---|---|---|---|---|---|---|---|---|---|---|
| Methods | gpt3.5 | palm2 | vicuna | gpt3.5 | palm2 | vicuna | gpt3.5 | palm2 | vicuna | gpt3.5 | palm2 | vicuna | gpt3.5 | palm2 | vicuna |
| **OpenAI detector** | 0.87 | 0.87 | 0.92 | 0.91 | 0.91 | 0.95 | 0.89 | 0.74 | 0.9 | 0.82 | 0.89 | 0.88 | 0.48 | 0.48 | 0.5 |
| | 0.91 | 0.94 | 0.92 | 0.69 | 0.96 | 0.81 | 0.86 | 0.87 | 0.93 | 0.6 | 0.95 | 0.82 | 0.97 | 1.0 | 0.98 |
| | **0.89** | **0.9** | **0.92** | 0.78 | **0.94** | **0.88** | **0.88** | 0.8 | **0.91** | 0.7 | **0.92** | 0.84 | 0.64 | 0.65 | 0.66 |
| **Roberta Large** | 1.0 | 1.0 | 1.0 | 0.72 | 0.9 | 0.88 | 0.99 | 0.99 | 0.99 | 0.89 | 0.93 | 0.92 | 0.97 | 0.99 | 0.99 |
| | **0.52** | 0.79 | 0.69 | **0.31** | 0.76 | 0.61 | **0.5** | 0.77 | 0.63 | 0.62 | 0.92 | 0.77 | **0.43** | 0.9 | 0.73 |
| | 0.68 | 0.88 | 0.82 | 0.43 | 0.82 | 0.72 | 0.66 | **0.86** | 0.77 | 0.73 | 0.92 | 0.84 | 0.6 | 0.94 | 0.84 |
| **Detect GPT** | 0.8 | 0.52 | 0.74 | 0.68 | 0.88 | 0.88 | 0.59 | 0.86 | 0.63 | 0.74 | 0.79 | 0.81 | 0.97 | 0.85 | 0.95 |
| | 0.85 | 0.65 | 0.91 | 1.0 | 0.97 | 0.97 | 0.67 | 0.7 | 0.65 | 0.83 | 0.81 | 0.89 | 0.89 | 1.0 | 1.0 |
| | 0.82 | 0.58 | 0.82 | 0.81 | 0.92 | 0.92 | 0.63 | 0.77 | 0.64 | **0.78** | 0.8 | **0.85** | **0.93** | **0.92** | **0.97** |
| **GPT Zero** | 0.6 | 0.49 | 0.62 | 0.91 | 0.89 | 0.93 | 0.81 | 0.78 | 0.81 | 0.57 | 0.58 | 0.58 | 0.62 | 0.58 | 0.61 |
| | 0.99 | 0.93 | 0.98 | 0.78 | 0.77 | 0.8 | 0.93 | 0.84 | 0.96 | 1.0 | 0.97 | 0.95 | 1.0 | 0.96 | 1.0 |
| | 0.75 | 0.64 | 0.76 | **0.84** | 0.83 | 0.86 | 0.87 | 0.81 | 0.88 | 0.73 | 0.73 | 0.72 | 0.77 | 0.73 | 0.76 |

Table 10: TT results on different datasets (Speech (S) or Interview (I)) for three LLMs: ChatGPT (gpt3.5), PaLM 2, and Vicuna-13B with **precision**, **recall**, and **F1 score** sequentially in each cell. The **bold scores** indicate the highest performing method (based on F1 score). The underlined scores indicate low precision scores (predicting most texts as AI-generated). The **bold & underlined scores** show a low recall score (can not detect most AI-generated texts).

**AA and AV performance is dataset specific:** Our study highlights significant performance variations in AA & AV across datasets, influenced by factors such as dataset type, domain, and modality. Daily-life conversation datasets (BASE, BNC, PAN) generally yield high F1 scores, except for MSU, which comprises simulated conversations with predefined topics, potentially limiting speakers' natural speaking styles. Conversely, talk-show-type datasets (USP, FTN) exhibit poor AA & AV performance due to a skewed distribution of samples from show hosts and the influence of Communication Accommodation Theory (Giles, 1973), suggesting speech adaptation to host styles.

**Individuals written & spoken samples are vastly different:** Our results affirm the distinctions between individuals' written and spoken texts, consistent with prior corpus-based research (Biber et al., 2000; Farahani et al., 2020). We also conduct an ablation study on SEC and PAN datasets, training on written texts only and testing on spoken texts (and vice versa). The results in Table 9 show a substantial decrease in the macro f1 score for both character n-gram and BERT-ft. Also, stylometry features exhibit poor performance, underscoring the stylistic differences between the two text forms and thus emphasizing the significance of separate authorship analysis for spoken text.

**AI-generated spoken and written texts have different characteristics:** Table 8 reveals a negative correlation between the Mauve score and the detection rate of LLMs, challenging the assumption that higher Mauve scores indicate harder-to-detect texts (Uchendu et al., 2023). Similarly, contrary to existing studies that highlight greater text diversity in humans compared to LLMs (Guo et al., 2023), we observe an opposite trend that humans tend to exhibit more repetitions in their speech. These findings suggest a further investigation into the specific characteristics of AI spoken language.

**How close are LLM-generated spoken texts to humans?** The AA-10(L) and AA-20() experiments reveal a decrease in the f1 score compared to AA with all human speakers, indicating that LLM-generated spoken texts exhibit greater similarity to other speakers in the experiments. It highlights the potential for further research in training LLMs to replicate individual spoken styles accurately.

**Which LLM is the winner?** Identifying the best LLM for spoken text generation remains an open question, influenced by multiple factors. PaLM2 demonstrates a lower GRUEN & DIV score and is more easily detectable than other LLMs in all datasets. On the contrary, ChatGPT exhibits lower Mauve scores, indicating its generation outside of human distribution, but its higher average recall value exhibits that it is difficult to detect.

# 6 Conclusion

We present HANSEN, a benchmark of human and AI-generated spoken text datasets for authorship analysis. Our preliminary analysis suggests that existing SOTA AA & AV methods behave similarly for spoken texts in most scenarios. However, the stylistics difference in spoken & written for the same individuals and AI-generated spoken texts show different characteristics than existing notions, emphasizing the need for a more nuanced understanding of the spoken text.

## Limitations

While the HANSEN benchmark encompasses multiple human and AI-generated spoken text datasets, it is essential to note that they are currently limited to English. Variations in spoken text structures and norms among humans differ significantly across languages (Crystal, 2007). Consequently, the performance of authorship analysis techniques may vary when applied to other languages (Halvani et al., 2016). Additionally, numerous spoken text datasets exist in various settings that are not yet included in the HANSEN benchmark. Furthermore, due to the computational demands of generating texts from LLMs, our study focuses on three specific LLMs while acknowledging the availability of other LLM models that could be explored in future research.

While our study focused on LLMs and their detection, it is important to acknowledge the ongoing arms race between the development of LLMs and LLM detectors (Mitchell et al., 2023). Therefore, our TT study may not encompass all LLM detectors, and it is possible that newly developed detectors outperform existing ones on these datasets. We deliberately refrained from finetuning any detectors on the spoken datasets as we believe that the evaluation of detectors should be conducted in open-ended scenarios. Although human evaluation of AI-generated texts still presents challenges (Clark et al., 2021), we consider it a crucial area for future work. Given the subjective nature of speech (Berkun, 2009), incorporating human evaluations can provide valuable insights into the quality of generated texts and further enhance our understanding of their performance in real-world applications.

## Ethics Statement

While the ultimate goal of this work is to build the first large-scale spoken texts benchmark, including LLM-generated texts, we understand that this dataset could be maliciously used. We evaluate this dataset with several SOTA AA, AV, and TT models and observe that there is room for improvement. However, we also observe that these findings could be used by malicious actors to improve the quality of LLM-generated spoken texts for harmful speech, in order to evade detection. Due to such reasons, we release this benchmark on huggingface's data repo to encourage researchers to build stronger and more robust detectors to mitigate such potential misuse. We also claim that by releasing this benchmark, other security applications can be discovered to mitigate other risks which LLMs pose. Finally, we believe that the benefits of this benchmark, outweighs the risks.

## Acknowledgements

This work was in part supported by U.S. National Science Foundation (NSF) awards #1820609, #1934782, #2114824, and #2131144, and by EU HE project SoBigData.eu receivinng funding from the European Union's Horizon 2020 research and innovation programme under grant agreements No. 654024, 871042, 101079043 and National Recovery and Resilience Plan - Prot. IR0000013 – Avviso n. 3264 del 28/12/2021, and by project PNRR - M4C2 - Investimento 1.3, Partenariato Esteso PE00000013 - "FAIR - Future Artificial Intelligence Research" - Spoke 1 "Human-centered AI", funded by the European Commission under the NextGeneration EU programme.

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

## A  HANSEN benchmark datasets & access

### A.1  Sources of new datasets

We have created three new datasets in our benchmark. The **SEC** dataset is compiled from the press release of the Security Exchange Commission (SEC)[2] website. It contains the transcripts of *speech* (spoken text) and *statements* (written text) from various personnel in the commission. We have created the Face the Nation (**FTN**) dataset by compiling the transcripts from the popular talk show program *Face the Nation*[3] by CBS News. Each episode is led by a host who discusses contemporary topics with multiple guests. We need to extract the speaker and align the corresponding utterance for a speaker from the raw transcripts. Finally, the **CEO** interview dataset contains interviews with different financial personnel, CEOs of companies, and other stock market associates. We have compiled this dataset by collecting the available public transcripts of these interviews from three different sources: CEO-today magazine[4], Wall Street Journal[5], and Seeking Alpha[6]. We have removed the questions from the interview hosts for authorship analysis purposes and only kept the answers provided by the hosts.

### A.2  Benchmark release & access

The datasets of HANSEN can be accessed through the Python Hugging Face library, as shown in Figure 5. In order to comply with the distribution rights of HANSEN datasets and tackle the existing challenges of AI-generated misinformation propagation, HANSEN datasets consist of three modules: (1) Open-source data/existing datasets that are free to re-distribute, (2) Open-source data that are public but may not be re-distributed, as users have to download/scrape themselves, and (3) AI-generated data that we have generated. Module (1) is accessible by default, (2) is partially included such that we only provide download-related info (e.g., URLs, line numbers, and Readme files), and scripts for downloading/scraping/preparing but do not contain the data, (3) is available by completing

```python
from datasets import load_dataset
import pandas as pd

# AA task on TED_small dataset
AA_TED= load_dataset(
    'HANSEN-REPO/HANSEN', name=
    'AA_TED_small', split='train')
df_AA = pd.DataFrame.from_dict(AA_TED)

# AV task on SEC dataset
AV_SEC = load_dataset(
    'HANSEN-REPO/HANSEN'
    , name='AV_SEC', split='test')
df_AV = pd.DataFrame.from_dict(AV_SEC)

# TT task on human vs ChatGPT on TED
TT_ChatGPT_TED = load_dataset(
    'HANSEN-REPO/HANSEN', name=
    'TT_ChatGPT_TED', split='train')
df_TT = pd.DataFrame.from_dict(
                TT_ChatGPT_TED)
```

Figure 5: Python code for loading HANSEN datasets from Hugging Face API

a "good-usage" form. We have also checked the existence of Personal Identifiable Information (PII) using off-the-shelf automated tools[7], for all three modules. While it found the full name, affiliation, and program/website contact information, it did not find any sensitive PII, such as identification numbers or personal contact information.

## B  Selection of subsets for LLM prompting

Due to the computational costs of generating text from LLMs, we chose to work with a subset of Hansen human datasets. First, we select the datasets with enough metadata and well-known speakers (such as TED speakers, SEC commissioners, or Tennis players). TED, Spotify, and SEC datasets are mostly monologues (speech category), and CEO & Tennis are interview datasets.

For TED datasets, we removed the talks with music or instrumental focused. Spotify and Tennis contain numerous samples in the original version. Therefore, we considered the top speakers with the most samples and used them for LLM generation. Also, for Spotify, we removed the samples where the speaker is not individual, such as a tutorial channel or multi-party collaborations. Similarly, for the CEO dataset, we considered the subsets from ceo-today magazines since the questions are specific and guest-focused rather than the overall

---

[2] https://www.sec.gov/news/speeches-statements
[3] https://www.cbsnews.com/face-the-nation/
[4] https://www.ceotodaymagazine.com/category/opinion/the-ceo-interview/
[5] https://www.wsj.com/pro/central-banking/topics/transcripts
[6] https://seekingalpha.com/author/ceo-interviews

[7] https://pypi.org/project/piianalyzer/0.1.0/,https://pypi.org/project/piicatcher/

financial situation like the subset in Wall Street Journal.

## C Prompts used for generating *spoken-text*

In our experiments, we explored different prompt techniques to enhance the coherence and semantic similarity of the generated content by LLMs. We observed that providing sufficient context with the prompt yielded better results. For the interview dataset categories, we included the original questions asked by the host. In the case of TED talks, we utilized the original talk description and the opening lines to allow the LLMs to learn the subtle style of the talks. We used the BART summarizer tool (Lewis et al., 2020) for the Spotify podcast and SEC speeches to obtain summaries, which were then used as prompts. Additionally, we explicitly instructed the LLMs to generate plain text to avoid any unwanted formatting. Table 11 shows the prompts used for different categories. In the case of speeches, we instructed the LLMs to generate speech content using the same word counts as the original speech or up to the maximum allowed number of tokens (typically around 1024 tokens or approximately 800 words for LLMs).

## D More about evaluating *spoken* nature of the AI-generated texts

In section 3, we provide a brief analysis to show a comparison of LLM-generated spoken vs. written (Table 3) in parallel with the human spoken vs. written and how the differences align with the humans. To further validate, we have performed a small-scale study on a subset (100 samples from each) of the LLM-generated spoken ($L_s$) and corresponding written ($L_w$) for the TED dataset and consider another human spoken ($H_s$) and written ($H_w$) in different domains ($H_w$ from Xsum (Narayan et al., 2018) datasets: news articles, $H_s$ is from BASE dataset in HANSEN: academic conversations) to show the domain independence. We measure the stylistic difference using the cosine distance of features in Table 10. We observe $dist(H_w, L_w) < dist(H_w, L_s)$ and $dist(H_s, L_s) < dist(H_s, L_w)$ , which can provide an overall idea that our LLM-generated *spoken* datasets contain more similarity with the *spoken* language rather than the written ones. Figure 6 portrays the distribution difference between some features in the written & spoken samples.

## E Experimental details for authorship analysis

The details about our AA, AV, and TT methods are discussed in Tables 13, 14, and 15. Since the development of new methods is not the primary purpose of our paper, we used them with the default configurations and hyper-parameter settings in most cases. For the text generations with LLM, we used the top_p = 0.7 to ensure more creativity as well as maintaining a coherent text and max_tokens to the highest limit for that LLM.

We applied a pre-processing step to ensure compatibility with various methods to achieve a uniform text length range for all authorship tasks. This involved removing samples from datasets that fell below-specified thresholds (100 for AA and AV, 200 for TT) as specific methods, such as Finetuned BERT, Adhominem, or OpenAI Detector, require specific text lengths. We sometimes combined multiple samples from the same conversation or interview to reach the desired length, as observed in datasets like MSU or CEO. Additionally, we employed sentence splitting for large samples such as TED or Spotify to create new samples with text lengths that remained within the maximum allowed token limits (approximately 1000 tokens) for different methods.

Table 16 shows the AA results for both small and large versions of the datasets. We consider large version N different for these datasets to ensure substantial samples per class for classification. We observe that performance drops more substantially for Transformer based methods when the number of speakers N increases. Therefore, it validates that DL methods underperform if per-class word counts decrease (Tyo et al., 2022). Also, character n-gram performs considerably better than word n-gram in all scenarios. Similarly, Table 17 shows the AV results with different metrics for all datasets. While we observe a similar trend for classifiers in both AA and AV tasks compared to written text, the overall performance of these methods is less than different written datasets, as observed in previous studies (Stamatatos, 2009; Neal et al., 2017; Abbasi et al., 2022; Tyo et al., 2022). This leaves room for further investigation regarding a more nuanced analysis of spoken texts.

We have run the experiments for each setup five times for the AA, AV, and TT tasks and reported the average in all tables. In most cases, we get the exact results for the methods for AA & AV, except

| Category | Prompt |
|---|---|
| **TED** | Generate a TED talk as <SPEAKER_NAME>, who is <SPEAKER_BIO>. The talk should be around <WORD_COUNT_RANGE>. The talk description is as follows: <TALK_DESCRIPTION> The original talk starts as follows: <FEW_STARTING_SENTENCES_FROM_ORIGINAL_TALK> |
| **Spotify** | Generate a Spotify podcast as <SPEAKER_NAME>around <WORD_COUNT_RANGE>words. Generated text should be plain text only. The original podcast summary is as follows: <SUMMARY_OF_THE_ORIGINAL_PODCAST> |
| **SEC** | Generate a speech as Security Exchange Commission person: <SPEAKER_NAME>about: <ORIGINAL_TITLE_OF_THE_SPEECH>. The speech should be around <WORD_COUNT_RANGE> words. The summary of the original speech is as follows: <SUMMARY_OF_THE_ORIGINAL_SPEECH> |
| **CEO** | Generate an interview with the guest <GUEST_NAME>. The following is the interview description: <INTERVIEW_DESCRIPTION>. The host will ask the following questions: <ORIGINAL_QUESTIONS_ASKED_BY_HOST> |
| **Tennis** | Create a post match interview with tennis player <PLAYER_NAME>who has <WON/LOST>the match against <OPPONENT_NAME>at <TOURNAMENT_STAGE>in <TOURNAMENT_NAME>. The reporter will ask exactly the following questions: <ORIGINAL_QUESTIONS_ASKED_TO_THE _PLAYER> |

Table 11: Prompts used for generating spoken texts in different categories

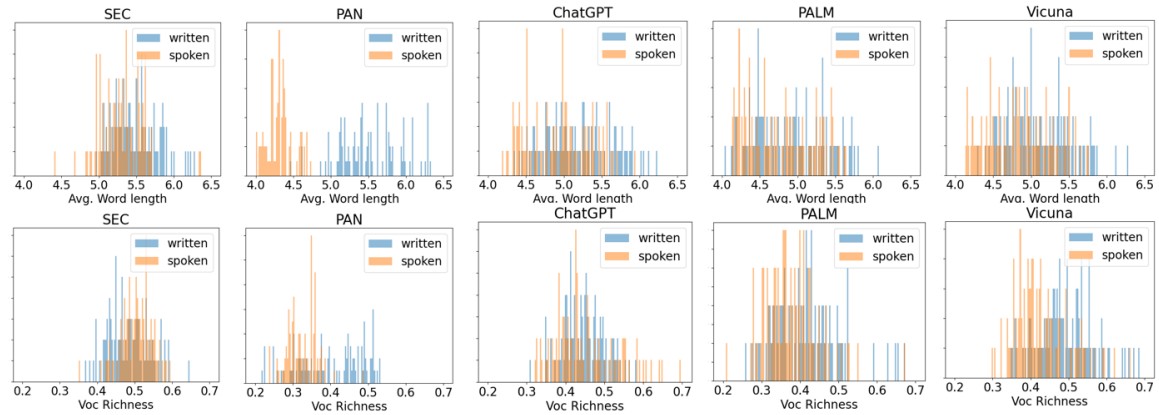

Figure 6: Avg. Word length and Vocabulary richness distribution of the different datasets for written and spoken text

| Method | Change of text property |
|---|---|
| **OpenAI detector** | No score or text property provided |
| **Roberta-Large** | Score for each label changes on average 0.007±0.0053 |
| **DetectGPT** | Z-score changes on average 0.003±0.0071 |
| **GPT-Zero** | Perplexity per line changes on average 1.05±0.087 Burstiness changes on average 3±4.27 |

Table 12: Effect of multiple runs on the TT tasks. Change of labels was not observed for any methods.

those that include random initialization, such as character n-gram for AV with grid search. However, the standard deviation of these results lies in the 0.0001-0.005 range, which will not impact

the overall result comparison. Although we observe minor fluctuations in the text property across different runs within a limited subset of our testing (e.g., z-score from DetectGPT or Perplexity from GPT-Zero), it is essential to note that the output, indicating the likelihood of AI/human origin, remains consistent across all our experimental scenarios. Table 12 summarizes the findings for the TT methods.

**A discussion about written and spoken texts of humans:** The overall dimension and differences between written and spoken language from statistical perspectives is a well-studied problem in various corpus linguistics (Biber, 1991; Biber et al., 2000; Brown et al., 1983). While there is a notice-

| AA method | Description |
|---|---|
| N-gram | We utilize the char, word level n-grams (represented as TF-IDF scores) with Logistic Regression (LR) classifier |
| Stylometry | We utilize the combination of both Linguistic Inquiry & Word Count (LIWC) (Pennebaker et al., 2001) and WritePrint (Abbasi and Chen, 2008) as stylometry features with LR classifier. LIWC analyzes the text based on over 60 categories representing various social, cognitive, and affective processes. WritePrint extracts lexical and syntactic features from the text, including char, word, letter, bigram, trigram, vocabulary richness, pos-tags, punctuation, and function words. |
| FastText+LSTM | We represent texts using FastText (Bojanowski et al., 2017) word embeddings since they can utilize the subword level information and train an LSTM model for the classification. |
| BERT-AA | Initially proposed by Fabien et al. (2020), it combines a pre-trained BERT with a dense layer for classification. |
| Finetuned BERT | As fine-tuned LM have benn SOTA in text classification tasks, we fine-tune BERT (*bert-base-cased*) on each dataset training set and evaluate it on the test set. |

Table 13: Description of the AA methods for spoken text.

| AV method | Description |
|---|---|
| N-gram | This method works as the PAN AV task baseline, where char n-grams represent each text. Cosine similarity is computed between text pairs, and threshold values $p_1$ and $p_2$ are optimized to adjust verification scores. |
| Prediction by Partial Matching (PPM) | Initially developed by (Teahan and Harper, 2003), it is another baseline for PAN AV tasks. Given a pair of texts, it estimates the cross-entropy of the second text using the first text's PPM model and vice versa. A LR model is then used to generate a verification score based on the mean and absolute difference of the two cross-entropies. |
| Feature differences | The method proposed by (Weerasinghe et al., 2021) achieved the second-highest performance in the PAN-2022 AV task. It utilized stylometric features, such as character and POS n-grams, special characters, function words, vocabulary richness, and unique spellings to represent text as feature vectors and used the a LR trained on the absolute difference between text pairs. |
| ADHOMINEM | Initially introduced by (Boenninghoff et al., 2019), this method is the highest-performing in recent PAN AV tasks (Kestemont et al., 2020). As the foundation for a Siamese network, this approach uses a hierarchical BiLSTM setup with Fasttext word embeddings and custom word embeddings learned using a character-level CNN. |
| Finetuned BERT | Similar to AA, we fine-tuned the BERT for each datasets training set with the concatenation of text pairs as sample. |

Table 14: Description of the AV methods for spoken text

able distinction between spoken and written text from a syntactic standpoint (Cleland and Pickering, 2006), whether an individual's subtle style is mirrored in both formats has yet to be answered computationally. Although the ablation study in our paper (on SEC and PAN datasets, Table 9) initially suggests no visible similarity, the result should not be considered conclusive. More parallel datasets for human individuals are needed to ensure conclusive evidence on whether individuals' unique style is represented in both forms. However, creating such parallel datasets manually will add unwanted bias. Therefore, it leaves room for future exploration as a separate study. Also, training LLMs to make them capable of replicating individuals' written and spoken styles may be a possibility that can shed more light on the problem.

**A discussion about TT task on AI-generated spoken text:** Although we observe the best-performing method in each category scores around 90% f1 in most scenarios, lower precision or recall value seems a significant problem for all detectors (Table 10). Also, recent evaluation studies on other benchmarks (He et al., 2023; Mitchell et al., 2023) (which are primarily written) show that the performance of DetectGPT or GPT Zero is very high (>80% on average); we do not observe such performance in spoken texts. Additionally, the overall result is less than the written counterpart. While fine-tuning language models for TT on these datasets would likely result in improved performance (He et al., 2023), relying solely on fine-tuning is not the ultimate solution. Therefore, future detectors should also consider the spoken text characteristics to make the detection more robust.

| TT method | Description |
|---|---|
| **OpenAI detector** | OpenAI developed this fine-tuned GPT model, which estimates the possibility of a text being AI-generated, specifically from the GPT family. We have accessed this detector from the official website (Kirchner et al., 2023) and considered "Possibly/Likely AI-generated" tags as AI-generated spoken text. |
| **Roberta -Large** | It was initially developed as the GPT-2 output detector model, obtained by fine-tuning a RoBERTa large model with the outputs of the 1.5B-parameter GPT-2 model (Conneau et al., 2020). |
| **DetectGPT** | DetecGPT (Mitchell et al., 2023) is a zero-shot AI-generated text classifier by generating perturbed samples from the original text and calculating their probability under the model parameters. |
| **GPT-Zero** | GPT-Zero (Tian, 2023) utilizes perplexity: to measure the complexity of text and Burstiness: to compare the variations of sentences to determine whether the text is AI-generated. |

Table 15: Summary of AI-generated spoken text detection methods

| | | | | | | | | *N = 10 speakers* | | | | | | | | |
|---|---|---|---|---|---|---|---|---|---|---|---|---|---|---|---|---|
| **Dataset** | TED | SEC | BASE | BNC | BNC14 | Tennis | Debate | MSU | BP | Court | Voxceleb | Voxpopuli | Spotify | USP | FTN | CEO | PAN |
| **Samples #** | 393 | 1208 | 4492 | 2066 | 1739 | 2915 | 2257 | 382 | 4715 | 29.4K | 1557 | 4341 | 42.2K | 2497 | 21.9K | 1338 | 2446 |
| **Char n-gram** | **0.95** | **0.9** | **0.98** | **0.89** | **0.92** | **0.99** | 0.83 | **0.33** | **0.98** | **0.91** | 0.86 | **0.94** | **0.98** | **0.52** | 0.58 | **0.85** | 1 |
| **Word n-gram** | 0.93 | 0.76 | 0.94 | 0.73 | 0.87 | 0.98 | 0.69 | 0.19 | **0.98** | 0.88 | 0.8 | 0.93 | 0.95 | 0.48 | 0.49 | 0.65 | 1 |
| **Stylometry** | 0.71 | 0.68 | 0.84 | 0.71 | 0.79 | 0.91 | 0.74 | 0.24 | 0.84 | 0.79 | 0.68 | 0.76 | 0.87 | 0.42 | 0.42 | 0.6 | 1 |
| **FastText** | 0.73 | 0.68 | 0.8 | 0.64 | 0.76 | 0.87 | 0.58 | **0.33** | 0.84 | 0.73 | 0.77 | 0.79 | 0.93 | 0.28 | 0.56 | 0.59 | 1 |
| **Bert-AA** | 0.69 | 0.55 | 0.94 | 0.7 | 0.66 | 0.76 | 0.72 | 0.16 | 0.69 | 0.82 | **0.87** | 0.87 | 0.9 | 0.3 | 0.46 | 0.72 | 1 |
| **Finetuned BERT** | 0.89 | 0.83 | 0.99 | 0.88 | 0.9 | 0.91 | **0.85** | 0.26 | 0.92 | 0.89 | **0.87** | 0.93 | **0.98** | 0.4 | **0.64** | 0.76 | 1 |
| | | | | | | *N = 30 for SEC, FTN, USP; N = 50 for TED, PAN; N = 100 for others* | | | | | | | | | | | |
| **Samples #** | 1219 | 1589 | 4492 | 10K | 6320 | 6848 | 4577 | 1978 | 15.4K | 70.7K | 12K | 17.7K | 145K | 282 | 33.7K | 3520 | 8544 |
| **Char n-gram** | **0.94** | 0.54 | **0.93** | 0.73 | **0.66** | **0.84** | **0.7** | **0.09** | **0.9** | **0.8** | **0.74** | 0.68 | **0.81** | **0.25** | **0.38** | 0.58 | 1 |
| **Word n-gram** | 0.82 | 0.43 | 0.82 | 0.48 | 0.46 | 0.72 | 0.51 | 0.06 | 0.86 | 0.68 | 0.59 | 0.67 | 0.56 | 0.14 | 0.34 | 0.35 | 1 |
| **Stylometry** | 0.64 | 0.46 | 0.64 | 0.54 | 0.51 | 0.66 | 0.45 | 0.04 | 0.56 | 0.42 | 0.43 | 0.39 | 0.61 | 0.24 | 0.27 | 0.35 | 1 |
| **FastText** | 0.6 | 0.42 | 0.61 | 0.43 | 0.37 | 0.46 | 0.34 | 0.05 | 0.5 | 0.28 | 0.38 | 0.33 | 0.47 | 0.18 | 0.26 | 0.28 | 1 |
| **Bert-AA** | 0.48 | 0.22 | 0.8 | 0.52 | 0.21 | 0.22 | 0.48 | 0.03 | 0.3 | 0.43 | 0.59 | 0.39 | 0.54 | 0.13 | 0.3 | 0.55 | 1 |
| **Finetuned BERT** | 0.75 | **0.63** | 0.84 | **0.78** | 0.56 | 0.56 | 0.64 | 0.04 | 0.63 | 0.77 | 0.66 | **0.77** | 0.77 | 0.16 | 0.32 | **0.62** | 1 |

Table 16: Results for the AA task for small and large number of speaker (N). The values indicate the Macro-F1 score for the proposed method. The **bold** and underlined values indicate the highest and second highest scores for specific datasets.

| **Dataset** | TED | SEC | BASE | BNC | BNC14 | Tennis | Debate | MSU | Parliament | Court | Voxceleb | Voxpopuli | Spotify | USP | FTN | CEO | PAN |
|---|---|---|---|---|---|---|---|---|---|---|---|---|---|---|---|---|---|
| **~Samples #** | 26.4K | 3244 | 16.4K | 22.8K | 18.2K | 15.3K | 19.4K | 6.4K | 20.9K | 29.4K | 28.9K | 21.1K | 29.7K | 10.1K | 13.4K | 7.9K | 17.8K |
| | | | | | | | | *AUC score* | | | | | | | | | |
| **N-gram** | 0.88 | 0.65 | 0.82 | 0.71 | 0.73 | 0.7 | 0.71 | 0.57 | 0.67 | 0.6 | 0.6 | 0.6 | 0.68 | 0.5 | 0.58 | 0.71 | 0.58 |
| **PPM** | 0.92 | 0.69 | 0.84 | 0.73 | 0.73 | 0.77 | 0.78 | **0.58** | 0.72 | 0.65 | 0.65 | 0.64 | 0.69 | 0.65 | 0.58 | 0.72 | 0.57 |
| **Feature difference** | 0.94 | 0.6 | 0.8 | 0.88 | **0.8** | 0.86 | 0.73 | 0.44 | 0.74 | 0.81 | 0.81 | 0.6 | 0.77 | 0.69 | 0.52 | 0.8 | 0.99 |
| **Adhominem** | 0.8 | **0.79** | 0.94 | **0.92** | 0.73 | **0.9** | 0.92 | 0.55 | **0.84** | **0.88** | **0.88** | 0.76 | 0.87 | 0.89 | 0.69 | 0.89 | **1.0** |
| **Finetuned BERT** | **0.96** | 0.69 | **0.98** | 0.89 | **0.8** | **0.9** | **0.97** | 0.52 | 0.83 | 0.87 | 0.87 | **0.8** | **0.9** | **0.9** | **0.72** | **0.94** | 0.56 |
| | | | | | | | | *F1 score* | | | | | | | | | |
| **N-gram** | 0.86 | 0.67 | 0.78 | 0.7 | 0.71 | 0.68 | 0.66 | **0.66** | 0.67 | 0.61 | 0.61 | 0.64 | 0.67 | 0.65 | 0.66 | 0.68 | 0.67 |
| **PPM** | 0.85 | 0.63 | 0.76 | 0.68 | 0.68 | 0.72 | 0.71 | 0.57 | 0.66 | 0.64 | 0.64 | 0.61 | 0.65 | 0.62 | 0.55 | 0.67 | 0.55 |
| **Feature difference** | 0.87 | 0.6 | 0.72 | 0.81 | 0.74 | 0.76 | 0.67 | 0.47 | 0.68 | 0.73 | 0.73 | 0.55 | 0.69 | 0.67 | 0.51 | 0.72 | 0.99 |
| **Adhominem** | 0.84 | **0.81** | **0.96** | **0.93** | **0.77** | **0.91** | **0.93** | 0.62 | **0.9** | **0.91** | **0.91** | **0.8** | **0.88** | **0.92** | **0.72** | **0.93** | **1.0** |
| **Finetuned BERT** | **0.91** | 0.64 | 0.93 | 0.79 | 0.69 | 0.81 | **0.91** | 0.46 | 0.74 | 0.79 | 0.79 | 0.73 | 0.83 | 0.81 | 0.67 | 0.88 | 0.67 |
| | | | | | | | | *Overall score* | | | | | | | | | |
| **N-gram** | 0.84 | 0.65 | 0.79 | 0.69 | 0.7 | 0.69 | 0.69 | **0.6** | 0.66 | 0.62 | 0.62 | 0.61 | 0.67 | 0.56 | 0.6 | 0.7 | 0.61 |
| **PPM** | 0.87 | 0.67 | 0.8 | 0.71 | 0.71 | 0.74 | 0.74 | **0.6** | 0.7 | 0.66 | 0.66 | 0.64 | 0.68 | 0.65 | 0.59 | 0.7 | 0.58 |
| **Feature difference** | 0.89 | 0.6 | 0.74 | 0.83 | **0.75** | 0.79 | 0.69 | 0.48 | 0.71 | 0.76 | 0.76 | 0.58 | 0.73 | 0.67 | 0.55 | 0.74 | **0.99** |
| **Adhominem** | 0.79 | **0.78** | 0.82 | **0.9** | 0.72 | **0.89** | 0.9 | 0.58 | **0.84** | **0.87** | **0.87** | **0.75** | **0.85** | **0.87** | 0.69 | **0.89** | **0.99** |
| **Finetuned BERT** | **0.92** | 0.67 | **0.94** | 0.82 | 0.73 | 0.83 | **0.92** | 0.54 | 0.77 | 0.81 | 0.81 | **0.75** | **0.85** | 0.83 | 0.69 | **0.89** | 0.62 |

Table 17: AUC, F1, and overall score (avg of all scores) for different methods in AV task.