# OpenReview forum: "HANSEN: Human and AI Spoken Text Benchmark for Authorship Analysis"
_EMNLP/2023/Conference — EMNLP 2023 Findings_

### Official Review · Reviewer_sdko · 2023-08-04

**Soundness:** 2

**Excitement:**

3: Ambivalent: It has merits (e.g., it reports state-of-the-art results, the idea is nice), but there are key weaknesses (e.g., it describes incremental work), and it can significantly benefit from another round of revision. However, I won't object to accepting it if my co-reviewers champion it.

**Paper Topic And Main Contributions:**

The paper presents the largest benchmark, HANSEN, which consists of a combination of human and AI-generated datasets for authorship analysis on spoken texts. Moreover, experiments with SOTA models are conducted to perform Authorship Attribution and Author Verification tasks on both human-spoken datasets and Human vs. AI spoken text detection tasks.

**Reasons To Accept:**

The paper is well organized, with detailed sections describing the proposed approach. Furthermore, the HANSEN benchmark proposed aims to contribute an effective way toward spoken text in terms of authorship analysis.

**Reasons To Reject:**

One thing that needs to be clarified is whether using AI-generated datasets to construct a benchmark is necessary (e.g.: the need to ensure the AI-text data is similar to the human text,...)

**Reproducibility:**

2: Would be hard pressed to reproduce the results. The contribution depends on data that are simply not available outside the author's institution or consortium; not enough details are provided.

**Reviewer Confidence:**

5: Positive that my evaluation is correct. I read the paper very carefully and I am very familiar with related work.

---

> ### Author Rebuttal · Authors · 2023-08-28
>
> > The paper is well organized, with detailed sections describing the proposed approach.
>
> We are delighted that the reviewer likes the organization of the paper.
>
> >One thing that needs to be clarified is whether using AI-generated datasets to construct a benchmark is necessary (e.g.: the need to ensure the AI-text data is similar to the human text,...)
>
> We are unsure about the clarification the reviewer pointed out in this sentence. However, We aim to address this aspect comprehensively, approaching it from two distinct perspectives.
>
> > …  whether using AI-generated datasets to construct a benchmark is necessary
>
>  We highlighted the significance of the research regarding AI-generated spoken text contents and their evaluation in the Introduction section of the paper, such as identifying synthetically generated scripts, detecting plagiarism, and solving potential copyright disputes (Lines 91-107). Therefore, creating and improving a benchmark with AI-generated spoken texts is necessary to initiate this research, especially in the age of AI and LLMs.
>
> > (e.g.: the need to ensure the AI-text data is similar to the human text,...)
>
> Also, ensuring the quality of AI text and how similar it is to the human text is crucial to assuring the quality of the benchmark. Recall that our work is preemptively studying an issue where AI can generate "spoken" text, much in the same way that ChatGPT generates written texts with human quality (that causes many technical and societal problems at the moment). Therefore, to study the characteristics of AI-generated spoken texts, we need to first ensure they are similar to human-generated spoken texts. In fact, this year's EMNLP theme track is the Large Language Model and the future of the NLP. Therefore, we belive that this paper can provide substantial merit to such future research.

---

### Official Review · Reviewer_Ky6M · 2023-08-07

**Soundness:** 4

**Excitement:**

4: Strong: This paper deepens the understanding of some phenomenon or lowers the barriers to an existing research direction.

**Paper Topic And Main Contributions:**

The authors introduce a new Benchmark for authorship analysis in human and automatically generated *Spoken Text*. The main contributions are as follow:
- The authors compile, pre-process and filter 17 existing human spoken datasets.
- The authors generate more than 20K samples of spoken text with 3 different LLMs: ChatGPT, PaLM2 and Vicuna.
- The authors extensively evaluate traditional authorship analysis methods on the benchmark resulting from the 2 previously mentioned contributions.

**Questions For The Authors:**

- Question A: What can you do in the short term to mitigate the risks that you mentioned in the Ethics Statement section.
- Question B: Do you have the rights to re-distribute all of these datasets? If so, under which license? Is there any Personal Identifiable Information (PII) in this dataset? If so, did you ask for consent from these people to redistribute their data?

**Reasons To Accept:**

- The authors curate and preprocess existing spoken datasets for this specific task, making them more accessible and usable.
- The authors generate new samples of automatically generated samples of spoken text.
- The authors provide extensive evaluation of existing AA methods on the proposed benchmark.
- The Benchmark and the evaluations as a whole constitute a valuable resource for the community
- The paper is clearly written.
- The authors provide a very valuable literature review in the introduction and related work sections.

**Reasons To Reject:**

- The Ethics Statement section leaves a lot to be desired, the authors claim that their data can be used by *”malicious actors”* in order to *”improve the quality LLM-generated spoken texts for harmful speech”*, yet they use this exact same claim to justify the release of the data. I.e. the authors do not provide any convincing arguments about what they can do to mitigate the harm that their own benchmark might cause.
- There is little to no discussion about the distribution rights that the authors hold to some of the curated datasets. For instance, there is no license information specially for the TED, CEO and FTN datasets. Yet the authors have already published this dataset in the Hugging Face Hub.

**Reproducibility:**

4: Could mostly reproduce the results, but there may be some variation because of sample variance or minor variations in their interpretation of the protocol or method.

**Reviewer Confidence:**

3: Pretty sure, but there's a chance I missed something. Although I have a good feel for this area in general, I did not carefully check the paper's details, e.g., the math, experimental design, or novelty.

---

> ### Author Rebuttal · Authors · 2023-08-28
>
> We thank the reviewer for highlighting these critical aspects of the paper. We try to incorporate these issues in the following points.
>
> > The Ethics Statement section leaves a lot to be desired, the authors claim that their data can be used by ”malicious actors” in order to ”improve the quality LLM-generated spoken texts for harmful speech”, yet they use this exact same claim to justify the release of the data. I.e. the authors do not provide any convincing arguments about what they can do to mitigate the harm that their own benchmark might cause.
>
> Thanks for pointing out this very important issue. What we really meant to state in the Ethics statement was the following. Like much research, our research can have both good and bad usage. While adversaries can use our dataset to harness the quality of AI-generated spoken texts for malicious usage, researchers can equally use our dataset to advance the science in generative AI for spoken texts and improve defensive solutions for good usage. More importantly, we believe the benefits of the latter outweigh those of the former, thus justifying our release of the dataset.
>
> As to the second point on mitigation, which is a very valid critique, we accordingly changed our data release protocol as follows:
> 1. Hansen will consist of three modules: (1) open-source data/existing datasets that we are free to re-distribute, (2) open-source data that we may not freely re-distribute but users have to download/scrape themselves, and (3) AI-generated data that we have generated.
>
> 2. Hansen will be re-packaged such that (1) is included by default, (2) is partially included such that we only provide download-related info (e.g., URLs, line numbers, and Readme files), and scripts that we built for the download/scraping/preparing, but do not contain the data, (3) is not included by default and will be provided to users who provide PI’s contact information and sign a “good-usage” form.
>
> We acknowledge that (3) cannot completely prevent Hansen from being used for malicious usage, but we hope to scare away those with bad intentions at least. That said, please understand that this issue of the “double-edged sword” usage of research works is not unique to Hansen but a general one for any research with a security flavor. We removed the existing anonymous release of Hansen in the Hugging Face and re-released it under the above protocol.
>
> > There is little to no discussion about the distribution rights that the authors hold to some of the curated datasets. For instance, there is no license information specially for the TED, CEO and FTN datasets. Yet the authors have already published this dataset in the Hugging Face Hub.
>
> Thanks for this point which we neglected to consider. Upon examining our release, we found that: [https://www.ted.com/about/our-organization/our-policies-terms/ted-talks-usage-policy] TED talks adopted the Creative Commons(CC) license, and others (such as FTN, CEO) assure the use for non-commercial purposes but prohibit redistribution. As such, we will repackage our release into three modules, explained above, and re-release a new version where module (2) contains only the download info of the data that prohibits redistribution of the actual copies.
>
> >  What can you do in the short term to mitigate the risks that you mentioned in the Ethics Statement section.
>
> To reduce the malicious usage of our dataset, instead of directly sharing our dataset with the general people in the Hugging Face, we will give access to those who (1) sign the “good-usage” form and (2) share the contact information of the PI. While this cannot completely prevent our dataset from malicious usage, we hope this will mitigate the short-term risks. We have removed the existing anonymous release of Hansen in the Hugging Face and will re-release it under the above protocol.
>
> > Do you have the rights to re-distribute all of these datasets? If so, under which license?
>
> We neglect not to consider the redistribution aspect of our newly created datasets (FTN, CEO). While they are publicly available and allow personal and educational usage, they prohibit redistribution of the original data. We will fix the problem using the three module approaches we proposed above.
>
> > Is there any Personal Identifiable Information (PII) in this dataset? If so, did you ask for consent from these people to redistribute their data?
>
> As mentioned above, we will repackage our data into three modules, where only modules (1) and (3) will have actual contents. In contrast, module (2), especially those we collected from web-scraping, such as FTN and CEO, has only download links (thus no data to include PIIs). We have checked the existence of PII using off-the-shelf automated tools, including (https://pypi.org/project/piianalyzer/0.1.0/) and (https://pypi.org/project/piicatcher/) for the module (1) and (3). While it found the full name, affiliation, and program/website contact information, it did not find any sensitive PII, such as identification numbers or personal contact information. Additionally, module (1) is openly available and accessible through other repositories; hence, it does not fall under our jurisdiction. Finally, module (3) is from AI-generated spoken text. Thus, data with PII appearance are from non-existent people or artificially generated.

---

### Official Review · Reviewer_bJGW · 2023-08-10

**Typos Grammar Style And Presentation Improvements:** 1. This is just my opinion, but it mi…
**Soundness:** 3

**Excitement:**

4: Strong: This paper deepens the understanding of some phenomenon or lowers the barriers to an existing research direction.

**Missing References:**

-

**Paper Topic And Main Contributions:**

This paper addresses the current research gap in authorship attribution, authorship verification, and AI-generated text detection, being primarily written texts, despite potential applications for spoken text analyses. The authors present the following contributions.
1. They curate existing speech corpora and new human-spoken datasets.
2. They generated spoken text datasets from three popular LLMs and demonstrated the data quality assessment.
3. They evaluate traditional authorship analysis methods on the resulting HANSEN benchmark datasets.


**Questions For The Authors:**

Question A: I wonder about the attribution of the authorship of spoken texts. For some formats, like TV shows, the speech comes from a combination of the pre-written script (not necessarily written by the speaker) and the speaker's ad-lib. With this property, would a speaker's speaking style be consistent across instances? Could false positives be possible (same speaker, different scriptwriter)?

Question B: Are the AA and AV experiments performed on the whole HANSEN benchmark datasets (i.e., the one including both human-spoken and A.I.) generated? If yes, do you label, for example, all ChatGPT-generated instances as a single class from a single entity? If yes, how do you ensure that generated texts from the same model retain the same persona or consistent speaking style?


**Reasons To Accept:**

1. The presented benchmark datasets introduce novel tasks of spoken text authorship analysis (AA, AV, and TT) on a large collection of diverse datasets in terms of speech contexts, formality levels, and topics.
2. This paper clearly demonstrates via experiments how baseline and SOTA methods of written AA, AV, and TT task perform on spoken text tasks.
3. The analysis section reveals that the spoken texts benchmark provides interesting research gaps for further research, such as that models trained on written texts do not perform well on spoken text test data, an observation worthy of further investigation.


**Reasons To Reject:**

1.  It is unclear in the main paper how the quality assessment of LLM-generated data and curation of existing datasets is performed. While the details are included in the appendix and briefly introduced in the benchmark creation section (like in Figure 4), it is essential (in my opinion) to discuss the process clearly. For example, one could question how to ensure these datasets exhibit different fundamental stylistic properties from written text ones like the PAN ones with lots of dialogue in fiction. One might also be skeptical of whether the LLM-generated dataset, despite the prompt, can be considered a spoken text and not just another written one. Together, all these might affect the applicability and novelty of the presented benchmark and the credibility of the experimental results and analyses presented.

2. It is unclear whether the experiment runs are performed once or multiple times. One may question the reliability of observations and analyses, with the possibility of some models performing better than others with the effect of randomness.


**Reproducibility:**

3: Could reproduce the results with some difficulty. The settings of parameters are underspecified or subjectively determined; the training/evaluation data are not widely available.

**Reviewer Confidence:**

3: Pretty sure, but there's a chance I missed something. Although I have a good feel for this area in general, I did not carefully check the paper's details, e.g., the math, experimental design, or novelty.

---

> ### Author Rebuttal · Authors · 2023-08-28
>
> We thank the reviewer for the detailed reviews and well-thought-out questions. We briefly discuss our response in the following points.
>
> > It is unclear in the main paper how the quality assessment of LLM-generated data and curation of existing datasets is performed.
>
> We agree with the reviewer that mentioning the curation and preprocessing steps of the existing datasets is essential since we are creating a benchmark. We briefly mention the procedures for creating new datasets in Appendix A. The source and brief details of the existing datasets are mentioned in Table 1. Due to the lack of space, we could not include this information and other preprocessing steps in the main paper. Given the extra page, we will discuss them in the revised main paper.
>
> > For example, one could question how to ensure these datasets exhibit different fundamental stylistic properties from written text ones like the PAN ones with lots of dialogue in fiction.
>
> Regarding the quality of the LLM-generated datasets, Figure 4 in Section 3.3 of the main paper highlights some quality scores that are widely used in evaluating the LLM-generated datasets [1,2,3]. We have also evaluated these datasets' Mauve score and readability score, which confirms the distribution similarity with the original human sources and the overall ease-to-read nature of spoken texts (similar found in Table 10 in the Appendix with human datasets). This will be included in the revised version/Appendix of the paper. Regarding the example of fiction, it can be written and spoken by nature [4]. The dialogue and conversations between characters are spoken language, while the description and other parts can be written. Therefore, comparing stylistic similarity between fiction and LLM-generated datasets in our paper might not be applicable.
>
> > One might also be skeptical of whether the LLM-generated dataset, despite the prompt, can be considered a spoken text and not just another written one.
>
> We agree that ensuring the spoken nature of LLM-generated text is crucial for the benchmark's validity. Therefore, in the Appendix section of the paper, we have shown a comparison of LLM-generated spoken vs. written (Table 10) in parallel with the human spoken vs. written, and the differences align with the humans. To further validate, we have performed a small-scale study on a subset of the LLM-generated spoken ($L_s$) and corresponding written ($L_w$) for the TED dataset and consider another human spoken ($H_s$) and written ($H_w$) in different domains (to show the domain independence) and measure the stylistic difference (using the features in Table 10). We observe $dist(H_w, L_w)<dist(H_w, L_s)$ and $dist(H_s, L_s)<dist(H_s, L_w)$, which can provide an overall idea that our LLM-generated datasets contain more similarity with the spoken language rather than the written ones.
>
> > It is unclear whether the experiment runs are performed once or multiple times. One may question the reliability of observations and analyses, with the possibility of some models performing better than others with the effect of randomness.
>
> Since we are creating a benchmark, train, test, and validation split for each dataset are fixed for further research. The papers that specifically mention the number of experiments run primarily consider cross-validation. We have run the experiments for each setup five times for the AA and AV tasks and reported the average in our paper. In most cases, we get the exact results for the methods, except those that include random initializations, such as char-n-gram AV with grid search. However, the standard deviation of these results lies in the 0.0001-0.005 range, which will not impact the overall result comparison. We will, however, report the number of experiments run in the revised version.
>
> Although we observe minor fluctuations in the text property across different runs within a limited subset of our testing (e.g., z-score from DetectGPT or Perplexity from GPT-zero), it is important to note that the output, indicating the likelihood of AI/human origin, remains consistent across all our experimental scenarios. The table below presents a summary of our rerun on the TED subset for the TT methods (five run per sample).
>
> | Method           | Change of text property (average over all samples)    | Comment                                                                           |
> |------------------|-------------------------------------------------------|-----------------------------------------------------------------------------------|
> |$\textbf{OpenAI detector}$   | N/A                                                   | As OpenAI pulled its AI Classifier, it is not possible to rerun the experiments.  |
> | $\textbf{Roberta Large}$    | Score for each label changes on average 0.007±0.0053  | Class assignment does not change                                                  |
> | $\textbf{DetectGPT }$       | Z-score changes on average 0.003±0.0071               | Class assignment does not change                                                  |
> | $\textbf{GPT Zero}$         | Perplexity per line changes on average 1.05±0.087, Burstiness changes on average 3±4.27    | Class assignment does not change
>
> > I wonder about the attribution of the authorship of spoken texts. For some formats, like TV shows, the speech comes from a combination of the pre-written script …
>
> We already had the same concern while building the benchmark, so we did not include movie/TV series subtitles as spoken text datasets in HANSEN since the speech will not be from the actual actors but from the scriptwriters. We also mention this issue for the SEC (scripted) and MSU (simulated) datasets as characteristics (Table 1) and how it can affect the result (Discussion section: Line 391-394). The first paragraph of Section A in the Appendix pointed out the criteria we have considered to make representative spoken text datasets. However, we thank the reviewer for pointing out that it could be an exciting study for future works to distinguish ad-lib from scripted part of the speech for TV/movies/other media.
>
> > Are the AA and AV experiments performed on the whole HANSEN benchmark datasets ...
>
> The AA experiment, involving human and AI-generated texts, contains top-7 human speakers and 3 LLMs (each LLM has been considered a speaker) simulating the 10-class AA problem [AA-10(L)] for each dataset. It was designed this way because we tried to simulate the scenario where LLM can perform as an individual speaker. However, we agree with the reviewer's point. Since texts from the same LLM are generated using a specific prompt (prompting them to symbolize the actual human persona), LLM is supposed to exhibit a different persona in all its instances. Therefore, we have also conducted an ablation study for the TED dataset with samples of the top 5 human speakers along with corresponding samples generated from the ChatGPT (considering five classes), Vicuna13B, and PaLM 2. So, it will be like a 20-class AA task (TED-20-w-LLM). The result comparison with the actual 20 class AA task (all human speakers, TED-20) is given in the following table (macro-f1 score). It demonstrates the substantial drop in performance, highlighting that LLMs can impersonate the actual human speaker.
>
> | Method       | TED-20  | TED-20-w-LLM  |
> |------------------|-------------|-------------------|
> | $\textbf{Char n gram}$  | 0.92        | 0.37              |
> | $\textbf{Stylometry }$ |  0.68       | 0.39              |
> | $\textbf{FastText}$ |  0.67       | 0.26              |
> | $\textbf{BertAA}$ |  0.56       | 0.46              |
> |$\textbf{Bert-ft}$  |  0.84       | 0.51              |
>
> > …  might be helpful to introduce character n-gram …
>
> We will update it in the revised version.
>
> > I don't think the full description of the dataset is necessary to include in the main text …
>
> We believe that at least a dataset description and characteristics in a short format are necessary to get a hold of the primary results presented in the main paper rather than going through the appendix all the time. Otherwise, it would be difficult to understand some aspects of the discussion and results, such as poor AA & AV performance for talk-show typed datasets FTN and USP (lines 395-400 in the discussion section). Also, sample and token size is essential to report to understand the size of the datasets and corresponding results for different methods. Understanding the curation and generation procedure without providing the dataset characteristics will be difficult for the readers, in our opinion.
>
> > In Table 8, it would be great to use non-color coding for accessibility reasons. You could use diacritics, underlining, or italics.
>
> We thank the reviewer for pointing out the issue. We will incorporate it in the revised version to ensure accessible reading.
>
>
> $\textbf{References}:$
>
> 1. Uchendu, A., Le, T., & Lee, D. (2023). Attribution and Obfuscation of Neural Text Authorship: A Data Mining Perspective. ACM SIGKDD Explorations Newsletter, 25(1), 1-18.
>
> 2. Pu, J., Sarwar, Z., Abdullah, S. M., Rehman, A., Kim, Y., Bhattacharya, P., ... & Viswanath, B. (2023, May). Deepfake text detection: Limitations and opportunities. In 2023 IEEE Symposium on Security and Privacy (SP) (pp. 1613-1630). IEEE.
>
> 3. Ji, Z., Lee, N., Frieske, R., Yu, T., Su, D., Xu, Y., ... & Fung, P. (2023). Survey of hallucination in natural language generation. ACM Computing Surveys, 55(12), 1-38.
>
> 4. Tannen, D. (1980, October). Spoken/written language and the oral/literate continuum. In Annual Meeting of the Berkeley Linguistics Society (Vol. 6, pp. 207-218).

---

### Meta-Review · Area_Chair_L99M · 2023-09-18

**Recommendation:** 4

**Metareview:**

This paper addresses the current research gap in authorship attribution, authorship verification, and AI-generated text detection, being primarily written texts, despite potential applications for spoken text analyses. The authors present the following contributions.

- They curate existing speech corpora and new human-spoken datasets.
- They generated spoken text datasets from three popular LLMs and demonstrated the data quality assessment.
- They evaluate traditional authorship analysis methods on the resulting HANSEN benchmark datasets.

Reasons To Accept:
- The presented benchmark datasets introduce novel tasks of spoken text authorship analysison a large collection of diverse datasets in terms of speech contexts, formality levels, and topics.
- This paper clearly demonstrates via experiments how baseline and SOTA methods perform on spoken text tasks.
- The analysis section reveals that the spoken texts benchmark provides interesting research gaps for further research, such as that models trained on written texts do not perform well on spoken text test data, an observation worthy of further investigation.
- The paper is clearly written.
- The authors provide a very valuable literature review in the introduction and related work sections.

Reasons To Reject:
- It is unclear in the main paper how the quality assessment of LLM-generated data and curation of existing datasets is performed.

---

### Decision · Program_Chairs · 2023-10-07

**Decision:**

Accept-Findings

**Comment:**

This paper addresses the current research gap in authorship attribution, authorship verification, and AI-generated text detection, being primarily written texts, despite potential applications for spoken text analyses. The authors present the following contributions.

- They curate existing speech corpora and new human-spoken datasets.
- They generated spoken text datasets from three popular LLMs and demonstrated the data quality assessment.
- They evaluate traditional authorship analysis methods on the resulting HANSEN benchmark datasets.

Reasons To Accept:
- The presented benchmark datasets introduce novel tasks of spoken text authorship analysison a large collection of diverse datasets in terms of speech contexts, formality levels, and topics.
- This paper clearly demonstrates via experiments how baseline and SOTA methods perform on spoken text tasks.
- The analysis section reveals that the spoken texts benchmark provides interesting research gaps for further research, such as that models trained on written texts do not perform well on spoken text test data, an observation worthy of further investigation.
- The paper is clearly written.
- The authors provide a very valuable literature review in the introduction and related work sections.

Reasons To Reject:
- It is unclear in the main paper how the quality assessment of LLM-generated data and curation of existing datasets is performed.